# Advances in Molecular Imaging for Neuroendocrine Neoplasms

**DOI:** 10.3390/cancers17122013

**Published:** 2025-06-17

**Authors:** Bradley Girod, Vikas Prasad

**Affiliations:** 1Department of Radiology, University of Texas Southwestern, 1801 Inwood Rd, Dallas, TX 75235, USA; 2Department of Radiology, Washington University School of Medicine, St. Louis, MO 63110, USA; pvikas@wustl.edu

**Keywords:** DOTATATE, DOTATOC, FDG, neuroendocrine neoplasms, PET, somatostatin receptors

## Abstract

In the past two decades, molecular imaging techniques, particularly PET/CT, have become the cornerstone modality for the diagnosis and staging of neuroendocrine neoplasms (NENs). These are a heterogenous and diverse group of neoplasms estimated to account for approximately 3–4 cases per 100,000 in the United States, but vary significantly around the world and by age demographics. Despite the diversity of these neoplasms, the vast majority express the somatostatin receptor (SSTR), which is why radiopharmaceuticals targeting somatostatin receptors have become the most commonly used ones for the diagnosis and staging of NENs. NENs that express low levels of the SSTR generally have poorer progressions and can be imaged with other radiopharmaceuticals, such as F-18-labeled fluorodeoxyglucose (FDG), to assess disease heterogeneity as well as the relative aggressiveness of the disease.

## 1. Introduction

Neuroendocrine neoplasms (NENs) represent a highly diverse group of tumors that can be functional or non-functional and are classified by their site of origin and pathologic appearance. These neoplasms vary from indolent to highly aggressive tumors, so their management is informed by the primary site, functionality of the tumor, extent of the metastatic disease, proliferation rate (Ki-67), and the clinical presentation of the patient. The incidence and detection rate of NENs have increased approximately 6.4-fold from 1973 (1.09 per 100,000) to 2012 (6.98 per 10,000), largely because of improved imaging techniques [1]. Somatostatin analogs, such as octreotide and lanreotide, are the first line of systemic therapy for the treatment of Grade 1 (G1) and Grade 2 (G2) neuroendocrine tumors (NETs). In the setting of localized disease, the therapy options include surgical excision; in the context of functional liver metastasis, loco-regional therapies, such as transarterial embolization or radioembolization, are available. In patients who have progressed clinically or radiographically, management includes chemotherapy, everolimus, sunitinib, cabozantinib, or peptide receptor radionuclide therapy (PRRT) [1,2,3]. The appropriate treatment algorithm is well-delineated for some patients; however, management becomes more complex when the disease is metastatic and heterogeneous. A single biopsied lesion often poorly represents the diversity of the systemic disease in patients with metastatic NENs, and even within a biopsied lesion there can be significant variability in tumor biology [4]. A sample primarily represents a subset of the tumor population in a lesion, and metastatic disease can have other properties that also change over time, making a true assessment of the grade difficult to determine. This is one of the primary reasons why PET/CT imaging with various agents is so important for disease staging and clinical decision making; numerous lesions can be functionally assessed in ways that were previously impossible. In the future, it is likely that multiple PET tracers will be utilized to fingerprint diseases in ways that were only previously possible in a pathology laboratory, leading to major advances in personalized medicine.

Prognostic and therapeutic success are measured in terms of the quality of life (QoL), progression-free survival (PFS), and overall survival (OS). The most robust measure of success has been OS, but in G1 and G2 NENs this endpoint can be difficult to achieve in prospective clinical trials because it is often many years before patients progress and the mortality endpoint is reached. Additionally, with advanced imaging techniques many more NENs are being identified than at earlier time points. Due to this reality and the recommendations coming from the National Cancer Institute’s Neuroendocrine Tumor Clinical Trials Planning Meeting in 2011, PFS has become the most important surrogate endpoint [5]. Since these recommendations were made, there have been important advances in therapies that can slow tumor progression, thus making this surrogate endpoint more meaningful. The connection between PFS and OS can be difficult to demonstrate for a single treatment option because of the long PFS durations, subsequent lines of therapy following progression, and crossovers to a different arm of a clinical trial [6,7,8]. Due to the difficulties and cost of performing prospective work to evaluate the long-term outcomes of an increasingly well-managed disease, the most valuable studies for assessing the OS may be multicenter observational cohorts. Frilling and colleagues evaluated the outcomes of patients with neuroendocrine tumors with liver metastases over the course of three decades, and found significant improvements in the overall survival among the more contemporary cohorts [9]. In more aggressive NENs such as small-cell lung cancer, there will be opportunities to demonstrate improvements in the OS; however, the field has become increasingly successful in treating many NEN tumor types, necessitating the use of progression-free survival in prospective trials. 

## 2. Current Practices in the Imaging of NENs

### 2.1. Somatostatin Receptor Imaging

Somatostatin is a peptide hormone that binds to somatostatin receptors (SSTRs) to down-regulate tumoral growth, assisting in disease control. These SSTRs, particularly SSTR2 expressed in neuroendocrine tumors, can be visualized noninvasively by labeling artificially synthesized somatostatin analogs with positron emitters, such as F-18, Ga-68, and Cu-64. Historically, gamma-emitting radiopharmaceuticals have been used, but they have become less commonly used due to their poorer performance than their positron-emitting counterparts. Indium-111 octreotide was used as a gamma emitter prior to the development of effective PET agents, but has been essentially replaced due to its inferior image quality and resolution. SSTR PET/CT has shown superiority over indium-111 octreotide and conventional diagnostic imaging, especially for identifying nodal metastasis and subtle disease [10]. While SSTR agonists are more often used, interest in SSTR antagonists has grown due to their increased tumor-to-background ratio and pharmacokinetic advantages. The current guidelines recommend performing an SSTR PET/CT for staging, following a therapeutic response, and for prognostication with standardized reporting procedures [11,12,13,14]. An SSTR PET/CT is critical for assessing the potential benefit of PRRT, local–regional therapies, and systemic treatment options. In the phase II prospective Lumen study, a Ga-68 DOATATATE PET/CT was shown to be an effective prognostic indicator, and patients with an SUV_max_ lower than 13.0 had a poorer response to PRRT [15]. Additionally, Ga-68 DOTATOC has been shown to be superior to CT and MR imaging for the initial staging of neuroendocrine neoplasms for patients being evaluated for surgical intervention, changing the management of 59.6% of cases in a 52-patient cohort study by Frilling and colleagues [16]. Changes in management may occur for various reasons, most commonly in the setting of newly identified metastases or disease heterogeneity not appreciated on tissue sampling. Therefore, while the pathological grading is important, some studies have shown no difference in the overall survival between G1 and G2, prompting Strosberg and colleagues to recommend a higher mitotic cutoff of five per high-power field rather than the current three per high-power field [17,18]. Additionally, sampling tissues from one to two sites of the disease or the surgical resection of the primary may not capture disease heterogeneity in patients with numerous lesions as effectively as a whole-body SSTR and FDG PET/CT.

### 2.2. Use of FDG-PET/CT in NENs

An FDG PET/CT plays an important role in the evaluation of neuroendocrine tumor heterogeneity, as it assists in the prognostication and assessment of candidates for PRRT [19]. Patients with high levels of dedifferentiated tumors and significant heterogeneity are more likely to have an increased FDG radiotracer uptake and glucose transporter type 1 (GLUT1) expression as their tumor moves towards a higher grade (Figure 1). This represents a transition from a well-differentiated neuroendocrine (NET) biology and tends to be less responsive to PRRT. While an FDG PET/CT is not considered routine in PRRT treatment evaluations, in selected subsets of patients it can be valuable. Many consider it an indispensable biomarker for pre-PRRT assessment, and this has prompted the development of various scoring systems, such as the NET PET by Chan and colleagues [19,20,21]. In lower-grade NENs, FDG is poorly sensitive, as low as 50%, which is anticipated in tumors with lower rates of glucose metabolism, making it poorly effective for staging low-grade NENs [22]. In these settings with lower-grade tumors, an FDG PET/CT can be highly impactful because it functions as an effective prognostic biomarker and whole-body mapping agent. To date, no prospective studies have demonstrated that an FDG PET/CT changes the clinical management, but many, including Binderup and colleagues, have recognized its value. They performed a prospective trial with a 10-year follow-up of 166 patients, of which 140 were either G1 or G2. These patients were assessed histologically with Ki-67 and with an FDG PET/CT; ultimately, the FDG PET/CT was the only independent predictor of the OS and PFS irrespective of the Ki-67 cutoff used in the multivariate analysis. It was also observed that patients with FDG-positive disease may exhibit a survival benefit when receiving PRRT; the median OS was 4.4 years versus 1.4 years in the patients not receiving PRRT [23]. These results indicate that there may be a subset of patients with FDG-positive disease who may benefit from PRRT to a greater degree than FDG-negative patients (Figure 2). While randomized controlled trials have not clearly defined the role of an FDG PET/CT for patient selection for PRRT, it serves as an important tool and the careful use of this biomarker can have important implications for patient care (Figure 3).

## 3. Novel Radiopharmaceuticals

### 3.1. SSTR Agonists

Multiple SSTR agonists are in use in current clinical practice, but other agents are being developed with various advantages. F-18 and Cu-64 exhibit more favorable half-lives than Ga-68, shorter positron ranges leading to potentially higher-resolution images, and central production sites yielding wider availability. There are several novel SSAs (somatostatin agonists), such as Cu-64 MeCOSar-Tyr^3^-octreotate (C-64 SARTATE), which have shown promising imaging characteristics in early studies and could be paired with Cu-67 SARTATE for therapeutic purposes [24]. F-18 AIF-NOTA-octreotide (F-18 AIF-OC) is another agent, which has a higher SSTR2 affinity compared to Ga-68 DOTA-TATE/NOC, and improved lesion detection rates of up to 91.1% compared to 75.3%, respectively [25]. Both SSAs F-18 AIF-OC and F-18 SIFAlin-TATE offer a higher tumor-to-background ratio (TBR) and a kit-like preparation, making them attractive products [26,27,28]. In a prospective study of 10 patients with histologically confirmed NENs who were assessed with both Ga-68 DOTATATE and F-18 AIF-OC, it was found that the lesion detection ratio was higher with F-18 AIF-OC, at 99.1%, versus 91.4% for Ga-68 DOTATATE. Of the 193 lesions identified by F-18 AIF-OC, 96.2% were confirmed by an MRI to be NEN lesions [27]. F-18 is often a desirable label for for radiopharmaceuticals because of its availability and favorable characteristics, making it popular for translation [28]. This field is observing a natural generational improvement over earlier agents like Ga-68 DOTATATE, and will further optimize SSTR agonists as other targets continue to be developed.

### 3.2. Radiopharmaceuticals Targeting SSTR Antagonists

The interest in SSTR antagonists has risen due to their superior pharmacokinetic profiles and TBR when compared to SSAs [29,30,31]. This is thought to be due to their higher binding affinity and the absence of the internalization of peptidomimetics [32]. However, it may be desirable for certain therapeutics to be internalized into cells; an optimal pharmacokinetic profile is also a potentially powerful therapeutic quality of beta-emitting agents. Moreover, the choice of chelator has an important effect on the binding affinity to SSTRs, with NODAGA-chelated agents exhibiting a 10-fold-higher affinity than DOTA-chelated agents [33]. This highlights the importance of chelator optimization in preclinical and clinical settings, which can be costly, especially for the creation of a novel radiopharmaceutical. For this reason, it is more practical that these optimizations be made naturally as part of a generational evolution; but, nevertheless, they should be considered early in the development of novel radiopharmaceuticals.

The most studied SSTR-2 antagonist is Ga-68 NODAGA-JR11 (Ga-68-OPS202), which has demonstrated higher sensitivity in lesion detection rates, especially in liver metastases; high TBRs; and a favorable biodistribution compared to Ga-68 DOTA TOC/TATE PET probes [34,35,36]. In a study performed by Lin and colleagues, Ga-68 NODAGA-JR11 was compared to Ga-68 DOTATATE and found to have an improved sensitivity of 91.7% (range, 87.6–95.7%) vs. 77.2% (range 71.0–83.4%), as well as a significantly better TBR in the liver [35]. If this promising early experience is validated, Ga-68 NODAGA-JR11 may become increasingly popular for clinical imaging and therapy, especially in patients with liver metastases.

When compared to Ga-68 DOTATATE, Ga-68 DOTA-JR11 showed a higher detection rate for liver metastases and an improved TBR, but lower detection for bone metastases [33]. The mechanism behind its lower performance in osseus lesions is not understood at this time. This antagonist also has minimal pituitary gland, adrenal gland, stomach, spleen, uncinate process, and small intestinal uptake, which may confer additional advantages to Ga-68 DOATATATE PET [31]. Some centers are beginning to develop significant experience with these novel antagonists, such as Liu and colleagues, who used Ga-68 NODAGA-LM3, Ga-68 DOTA-LM3, Ga-68 NODAGA-JR11, and Ga-68 DOTA-JR11 in 549 patients and found that these tracers performed favorably to an SSA. The patient-level sensitivity, specificity, and accuracy were 91.0% (443/487), 91.9% (57/62), and 91.1% (500/549), respectively. In the study, Ga-68 NODAGA-LM3 was superior in terms of the highest SUV of the most avid lesion, and was considered to have the most favorable imaging profile when evaluating the tumor-to-liver ratio (TLR) and SUVs [37].

### 3.3. Non-SSTR Radiopharmaceuticals

SSTR targeting is expected to remain the cornerstone of NEN imaging; however, exploring alternatives may prove valuable for imaging and treating NEN subtypes or for characterizing heterogeneous disease (Table 1). It is possible that in the coming years multi-tracer imaging and multiplex PET/CT will become increasingly popular for imaging and treating multiple tumor types simultaneously [38]. Multiplex PET/CT is beyond the scope of this discussion, but it allows for multiple radiopharmaceuticals to be visualized simultaneously by utilizing prompt gamma emissions that co-occur with the annihilation events.

**Table 1 cancers-17-02013-t001:** Various radiopharmaceuticals used in imaging of neuroendocrine neoplasia: their class, targets, and brief description of uses and properties.

Radiopharmaceuticals	Radioisotopes	Class	Target	Description
Ga-68 DOTA-TATE, DOTA-NOC, DOTA-TOC	Ga-68 short half-life. Generator produced.	SSTR agonist	SSTR	Widely used.
Cu-64 DOTA-TATE	Cu-64 longer half-life. Cyclotron produced.	SSTR agonist	SSTR	Widely used. Longer half-life than Ga-68. Lower photon energy than Ga-68.
F-18 Alf-NOTA-octreotide	F-18 intermediate half-life. Cyclotron produced.	SSTR agonist	SSTR	Higher SSTR2 affinity than Ga-68 DOTA-TATE/NOC.
F-18 SiFAlin TATE		SSTR agonist	SSTR	Higher tumor-to-hepatic ratio than Ga-68 DOTA-TATE/NOC. On-site labeling kit.
Cu-64 SAR-TATE		SSTR agonist	SSTR	Theranostic pair with Cu-67. Lower photon energy and longer half-life than Ga-68 DOTA-TATE/NOC.
Ga-68 DOTA-JR11		SSTR antagonist	SSTR	Higher number of receptor binding sites and TBR than Ga-68 DOTA TATE/NOC. Accumulates in tumors. Therapy pair with Lu-177 DOTA-JR11.
Ga-68 DOTA-LM3		SSTR antagonist	SSTR	Higher number of receptor binding sites. Higher TBR than Ga-68 DOTA-TATE/NOC. Accumulates in tumors. Therapy pair with Lu-177 DOTA-LM3.
F-18 FDG		Glucose uptake	GLUT 1	High grade nets. Heterogeneous disease. Contraindication to PRRT.
Ga-68 FAPI		Fibroblast-activated protein	Fibroblast-activated protein inhibitor	Uptake in many different cancers. Therapy potential.
Ga-68 CXCR4 (Ga-68 PentixaFor)		Chemokine receptor	C-X-C motif chemokine receptor	Dedifferentiated NET. Index of aggressiveness. Similar prognostic indicator to FDG.
Ga-68 DOTA-CCK-66		G protein coupled	Cholecystokinin 2 receptor	Pan NET agent. Renal toxicity.
Ga-68 or F-18 DOTA-Exendin-4		GLP1 R expression	GLP1 receptor	Insulinoma. Renal toxicity. High sensitivity compared to SSTR and MRI.
F-18 MFBG		Norepinephrine	Norepinephrine receptor	Benefits of F-18 labeling over I-123 MIBG. Neuroblastoma, paraganglioma, pheomchromocytoma.

Legend: SSTR, somatostatin receptor; TBR, tumor-to-background ratio; GLUT1, glucose transporter 1; CCK2, cholecystokinin receptor 2; GLP1, glucagon-like peptide 1; MFBG, meta-flouro-benzylguanadine [10,39,40].

Fibroblast activating protein inhibitor (FAPI) tracers have been drawing considerable interest in academic centers over the past decade as a tool for imaging multiple cancer types. It is well-known to the medical community that among malignant cells there are fibroblasts, immune cells, extracellular matrices, and endothelial cells in the tumor microenvironment. Of these, FAPI targets cancer-associated fibroblasts (CAFs) that play an important role in facilitating the production of extracellular matrix proteins, the synthesis of ligands and growth factors, immune evasion, and therapeutic resistance. In the setting of neuroendocrine tumors, FAPI tracers may be able to image dedifferentiated disease and SSTR-negative NENs, but will be forced to compete with other agents for this position [41,42]. F-18 FDG is currently an important prognostic indicator because it highlights metabolic activity within tumor environments. If FAPI were shown to have a prognostic benefit in addition to identifying SSTR-negative lesions, it may find a place in the staging and restaging of NENs. Therapy with FAPI tracers is unlikely to subvert the role of SSTR agonists and antagonists; however, it remains unexplored and may play a role in poorly differentiated disease [43].

Ga-68 CXCR4 (Ga-68 PentixaFor) binds to a C-X-C chemokine receptor (CXCR4), which is often overexpressed in poorly differentiated NETs, also referred to as neuroendocrine carcinomas (NECs). These tumors are dedifferentiated and often do not robustly express SSTR, necessitating the use of other radiopharmaceuticals, which has historically been FDG, but CXCR4-targeting agents may come to play an important role if therapeutics can be linked to the expression of this chemokine receptor [44]. A study conducted on 12 patients using Ga-68 PentixaFor, FDG, and Ga-68 DOTATOC showed that the role of PentixaFor in evaluating well-differentiated NETs is limited; however, as the tumor biology became more poorly differentiated, PentixaFor performed better. Therefore, PentixaFor may serve as a noninvasive imaging tool for evaluating CXCR4 endo-radiotherapy in dedifferentiated SSTR-negative NETs [45,46].

Cholecystokinin receptor 2 is a promising target that may serve as a pan NET agent, but is proving to be an interesting tracer for medullary thyroid cancer imaging. While more work must be conducted to further understand its role, Ga-68 DOTA-CCK-66 was found to have a similar imaging performance as F-18 DOPA in a patient with medullary thyroid carcinoma [47]. Preclinical work is being performed to evaluate the effectiveness of Ga-68 DOTA-CCK-66 in the setting of medullary thyroid cancer with different radio-labeled isotopes. Viering and colleagues have also utilized Ga-68 DOTA-CCK-66 in SCLC that demonstrated low SSTR 2 expression [48]. Various applications may exist, but currently medullary thyroid cancer is being explored with interest.

F-18 F-DOPA is a L-DOPA analog, a compound that is used to synthesize multiple neurotransmitters, such as dopamine, norepinephrine, and epinephrine. Its potential as a NET imaging agent stems from its transportation by the neutral amino acid transporter LAT/4 F2 hc, which is coupled to the mammalian target rapamycin (mTOR) signaling pathway [49]. This different mechanism may confer advantages, and several studies have shown an increase in the number of lesions detected when compared to Ga-68 DOTATOC. Ouvard and colleagues performed a comparison of 41 patients with ileal NETs and found that F-18 F-DOPA showed a similar detection rate of 97%; however, in the per-lesion analysis, F-18 F-DOPA performed better than Ga-68 DOTATOC, with an overall detection rate of 96% compared to 80%; *p* > 0.001. More metastatic lesions were detected using F-18 F-DOPA, making this a potentially promising tracer for primary tumor detection and preoperative staging [50]. Veenstra and colleagues observed that in patients with increased 5-HIAA greater than 200 nmol/L, serotonin greater than 20 nmol/L, and chromogrannin A greater than 185 ug/L, there were improved lesion detection rates [51]. While this must be further explored, it may be leveraged in specific clinical scenarios for improved cancer staging and serve as a complement to SSTR imaging.

F-18-Meta-flourobenzylguanidine (F-18 MFBG) is a radiotracer of interest for the evaluation of neuroblastomas, paragangliomas, pheochromocytomas, and extra-adrenal paragangliomas without SDHx mutations [52]. In particular situations, F-18 MFBG has shown superior performance to MRI and Ga-68 DOATATOC PET/CT, for example, in a patient with recurrent pheochromocytoma [53]. In a pilot study, using F-18 MFBG in children with a long-axial field of view was not only superior in terms of the imaging performance, but also yielded shorter scan times, a higher sensitivity, and resulted in less sedation and general anesthesia [54]. Wang and colleagues utilized F-18 MFBG PET/CT as an alternative to Ga-68 DOTATATE PET/CT in metastatic pheochromocytoma and paragangliomas. F-18 MFBG detected lesions in all the patients, while Ga-68 DOTATATE detected lesions in 27 (96.4%) of the patients. In addition, a total of 686 lesions were detected by both tracers, with 33 uniquely positive lesions in the MFBG group and 16 uniquely DOTATATE detected lesions [55].

Ga-68 DOTA-Exendin-4 and F-18 exendin-4 are glucagon-like peptide-1 receptor analogues that show high diagnostic accuracy for benign insulinomas, but their lower expression is seen in malignant insulinomas. Boss and colleagues investigated an exendin PET/CT in 69 adults with biochemically proven adult endogenous hyperinsulinemic hypoglycemia, and found that an exendin PET/CT had an accuracy of 94.4% (95% CI, 84.6–98.8), which was greater than that of a DOTA-SSA PET/CT (64.8%; 95% CI, 50.6–77.3%), contrast-enhanced CT/contrast-enhanced diffusion-weighted-imaging MRI (83.3%; 95% CI, 70.7–92.1%), and endoscopic ultrasound (82.8%; 95% CI, 64.1–94.1%) [56]. These agents may be helpful for determining whether surgical or medical therapy is appropriate. Additionally, because GLP-1 agonists are widely used for metabolic syndromes, there may be broader applications for these tracers beyond identifying NENs. Metabolic disease may be an area of interest for these tracers and may even occur prior to the development of NENs, but that is beyond the scope of this discussion.

### 3.4. PET/MR

The development of new diagnostic radiopharmaceuticals is changing patient care, especially SSTR imaging; however, a PET/MRI may confer additional advantages for the staging and evaluation of liver metastases, post-treatment changes, and better lesional characterization. This tool has been used in academic centers for a number of years, but it only became commercially available around 2010. Most diagnostic workflows in the United States consist of a patient having a PET/CT followed by an MRI when deemed necessary; however, there is growing interest in using a PET/MRI as a “one stop shop” for numerous types of malignancies. This is particularly appealing in pediatrics, where limiting the amount of ionizing radiation is thought to be more important for reducing the risk of future malignancies. While it is difficult to demonstrate the need for simultaneous PET and MR images, their utility, if available to a patient, is exciting as well as potentially beneficial.

There are three manufacturers of PET/MRI machines that are available for medical use in the United States: SIGNA (GE Healthcare), uPMR 790 (United Imaging), and Biograph mMR (Siemens) [57]. These systems utilize a T 3 magnet and a lutetium scintillator to acquire PET and MR images simultaneously, unlike PET/CT systems, which acquire the images in series. A collaborative approach is important for establishing a successful PET/MR imaging center, due to the complexity of image acquisition, quality control, and patient care. While the acquisition parameters vary by institution, the most widely utilized protocol is 2 min of data collection per bed position. High-quality co-registration can then be achieved due to the advances in technical respiratory gating and motion artifact correction that are superior to those of PET/CT [58,59]. Motion correction is essential for intra-abdominal imaging near the diaphragm, because while patients may breathe freely during PET acquisitions, there are breath-holding techniques conducted during some MR sequence acquisitions. Having well-established motion algorithms is critical for NEN imaging, because the most common site of metastases is the liver, and a PET/MR is highly capable of producing high-quality co-registration if performed appropriately. There are an increasing number of motion correction operations available that minimize motion-related image artifacts, partial volume averaging, and facilitate high-quality image acquisition [60,61,62,63]. These previously highlighted technical challenges are of the most relevance to NEN imaging, and these challenges are effectively managed at experienced centers.

Several prospective studies have examined the role of PET/MR in the diagnostic work-up of NENs, reporting high sensitivity and specificity when compared to PET/CT. Schreiter et al. compared Ga-68 DOTATOC PET/MR and Ga-68 DOTATOC PET/CT in a twenty-two consecutive patient cohort with suspected liver metastases, and found that the sensitivity of the PET/MRI was 91.2% (95% CI, 84.3–95.7%) and its specificity was 95.6% (95% CI, 87.6–99.1%) compared to the PET/CT, which were 73.5% (95% CI, 64.3–81.3%) and 88.2% (95% CI, 78.6–99.1%) [64]. Additionally, a study performed by Berzaczy and colleagues also compared Ga-68 DOTANOC PET/MRI and Ga-68 DOTANOC PET/CT and found that the sensitivity of the PET/MRI was 90.8% (95% CI, 77.8–96.6%) and its specificity was 100% (95% CI, 97–100%) compared to those of the PET/CT, at 81.6% (95% CI, 68–91.2%) and 100% (95% CI, 97–100%), respectively [65]. While it is encouraging that experienced PET/MRI centers have been able to leverage its potential advantages, it will not replace PET/CT due to the poor availability and cost. However, in some clinical scenarios PET/MRI may be prioritized to address a patient’s needs; for example, in the study performed by Schreiter and colleagues, hepatic metastases were suspected. A PET/MRI offers clear advantages with respect to improved image co-registration and the detection of hepatic metastases in NENs. The protocols can also be streamlined to focus on the most important anatomy, and when whole-body examinations are performed, a dedicated liver and pancreas protocol can then be acquired. Additionally, an abbreviated metastases-focused protocol may be performed, including only diffusion-weighted imaging and hepatobiliary phase gadoxetic acid-enhanced T1-weighted MRI [66]. This aims to decrease the overall exam time, improve patient tolerance, and can be performed with respiratory gating for near-perfect image co-registration [59]. As the field continues to develop and PET/MRI becomes more available, it may be increasingly used for the staging and restaging of NENs, as it enables the increased detection of new lesions, lesional growth, and superior planning for PRRT, owing to the strengths in the anatomic and molecular imaging qualities of the exam [67].

To summarize, SSTR PET/MRI provides a single solution for NEN imaging, with superior sensitivity, specificity, and soft-tissue contrast. Soft-tissue contrast is especially valuable in liver imaging, which is the most common site of distant metastases. It is likely that further optimization will allow for more rapid image acquisition and reduced artifactual image degradation. Finally, PET/MRI saves young adults and pediatric patients from radiation when serial restaging examinations are required [68].

## 4. Follow-Up of Patients with NEN and Future Directions

The follow-up of patients with NEN is often complicated by the high heterogeneity of tumor evolution as well as unclear symptoms [69]. Patients with G1 and G2 NETs are often followed-up by CT/MRI at 3–6 month intervals, whereas SSTR PET/CT is generally used only yearly, and sometimes at even longer intervals. In addition, wherever suitable, the chromogranin A and serotonin levels in the blood or the 5-HIAA in 24 h urine are also used for the follow-up of NEN patients. For G3 NET as well as NEC patients, the frequency of follow-up is shorter and driven by the tumor characteristics. There are some novel biomarkers, like NETest2.0, that may be useful, e.g., for ordering imaging when there is clear evidence of disease progression (Figure 4) [70]. With the advancements in image segmentation tools, some of which are AI-driven, it may be possible to better characterize tumor heterogeneity and total tumor volume, which, in turn, may help with the response prediction and prognostication of various systemic therapies for NENs [71].

## 5. Conclusions

There have been considerable improvements in the science, diagnostics, and therapeutic efforts focused on these increasingly recognized neuroendocrine neoplastic processes. Improvements in radiopharmaceuticals and PET imaging, particularly SSTR imaging, have changed what is possible and are guiding the treatment of NENs. Optimal staging with PET imaging has allowed surgeons to operate with confidence, select liver-directed therapies in the appropriate setting, and give PRRT or medical therapy when needed. PET/MR represents an exciting opportunity to stage disease and plan therapy with a level of precision not previously possible in a single imaging solution. PET/MR offers improved soft-tissue contrast, superior co-registration, and molecular characterization of target lesions, particularly in the liver. The use of multiple radiopharmaceuticals imaged serially, or simultaneously as in multiplex PET imaging, may develop to become an important tool for the improved molecular characterization of heterogeneous disease. More molecular information will allow clinicians to fingerprint patients, leading to improved prognostic information as well as therapeutic choices.

To move the field forward using these tools, the focus must be on optimally treating each patient in an individualized way. For this to occur, it is imperative that genetic and molecular tumor profiling improve, which will depend heavily on radiopharmaceutical development. This can be achieved by leveraging new radiopharmaceuticals, improving pathologic sampling and assessment, developing new and refining current biomarkers, and quantitative machine-driven analysis. These refinements will allow for improvements in treating neuroendocrine tumors that have historically been difficult. As these technologies continue to evolve, they will allow physicians to prognosticate more accurately, deliver the correct therapy at the appropriate time, and improve patient outcomes, ultimately moving towards an ideally personalized care.

## Figures and Tables

**Figure 1 cancers-17-02013-f001:**
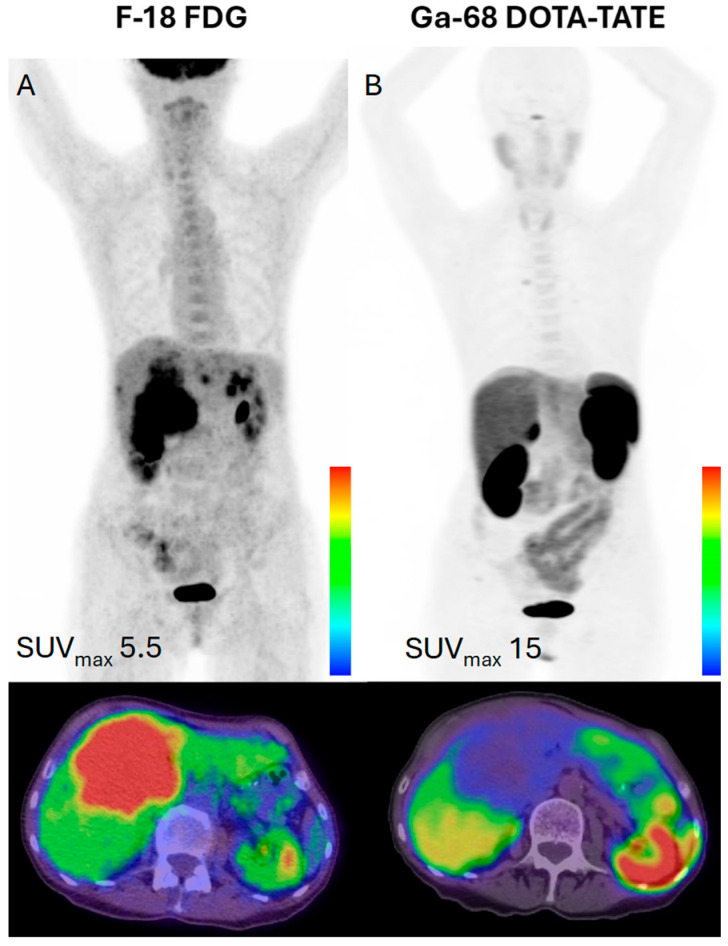
A 54-year-old female with a G2 (Ki-67.5%) neuroendocrine tumor of the pancreatic body with metastasis to a lymph node adjacent to the pulmonary ligament. (**A**) The F-18 FDG PET/CT MIP (maximum image projection) and fused axial FDG PET/CT image show the expected physiologic uptake in the brain, gastrointestinal and genitourinary tracts, and heart, with no FDG-avid disease in the chest. (**B**) The Ga-68 DOTA-TATE PET/CT shows physiologic tracer distribution in the pituitary, thyroid, liver, spleen, and gastrointestinal and genitourinary tracts, with an intensely radiotracer-avid focus adjacent to the pulmonary ligament, consistent with metastatic disease. The MIP images were scaled to an SUVmax value of 5.5 for the FDG PET/CT and 15.0 for the DOTATATE PET/CT normalized to the liver background. The images were acquired at UT Southwestern Medical Center.

**Figure 2 cancers-17-02013-f002:**
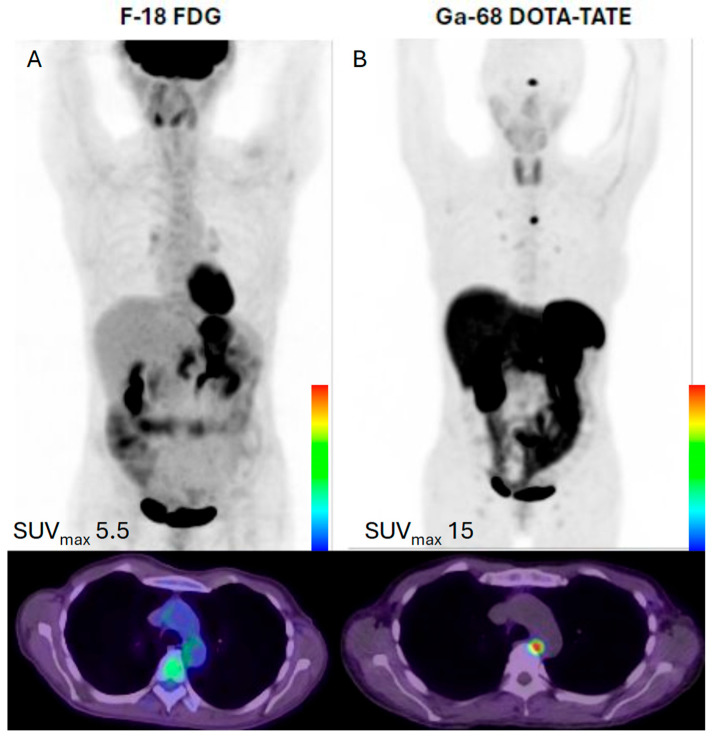
A 70-year-old female with poorly differentiated neuroendocrine carcinoma, G3 (Ki-67 80–90%). (**A**) The F-18 FDG PET/CT maximum intensity projection (MIP) images and fused axial PET/CT image show physiologic tracer distribution in the brain and gastrointestinal and genitourinary tracts, with an intensely radiotracer-avid mass spanning multiple hepatic segments, primarily in segments 5 and 8. (**B**) The Ga-68 DOTA-TATE PET/CT physiologic distribution in the pituitary, thyroid, liver, spleen, and gastrointestinal and genitourinary tracts, with an area of relative photopenia in the location of the FDG-avid hepatic mass. There can be several reasons for the lack of somatostatin receptor expression in the markedly FDG-avid hepatic lesions, such as the dedifferentiated tumors losing somatostatin receptor expression during tumor evolution (under therapeutic pressure) or the primarily aggressive clones may be de novo somatostatin receptor negative. The MIP images were scaled to an SUVmax value of 5.5 for the FDG and 15.0 for the DOTATATE PET/CT normalized to the liver background. The images were acquired at UT Southwestern Medical Center.

**Figure 3 cancers-17-02013-f003:**
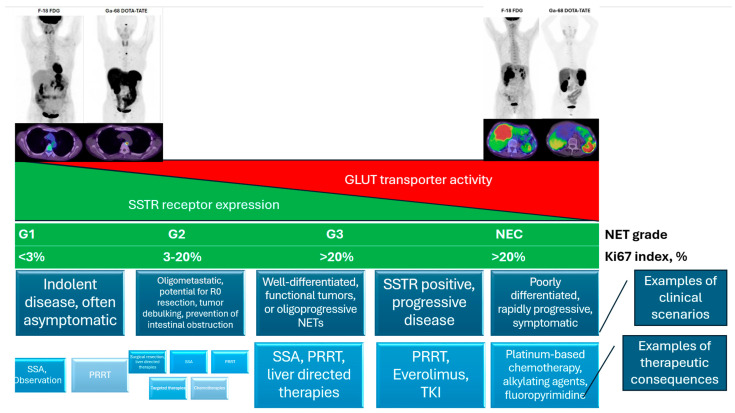
A summary of the relative tumor grades based on the WHO classification, general characteristic radiotracer uptake by the Ki-67 index, common treatment scenarios, and guideline-based treatment options. The previously shown images are given as examples representing opposite ends of the disease spectrum that are be commonly observed in clinical practice. The patient in the right upper panel demonstrates a well-differentiated NET with low FDG uptake in the nodal metastasis adjacent to the pulmonary ligament. The patient in the upper-left panel demonstrates an intensely FDG-avid mass in the liver without evidence of significant SSTR expression.

**Figure 4 cancers-17-02013-f004:**
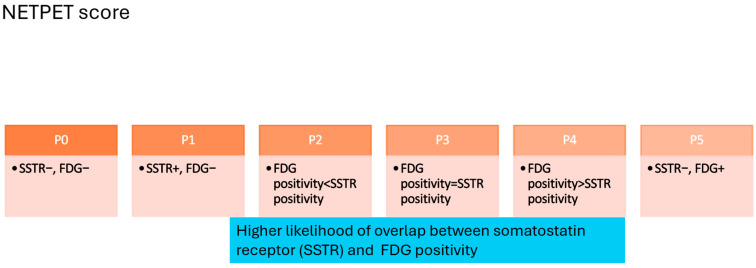
Summary of the NETPET scoring system, which highlights the variable radiotracer uptake based on the radiomic features of NENs. P0 represents a lesion with no SSTR or FDG radiotracer positivity. P1 lesions demonstrate an SSTR-targeted radiotracer positivity without significant FDG radiotracer positivity. P2–P4 lesions express variable SSTR positivity. P5 represents a poorly differentiated neuroendocrine tumor that no longer expresses detectable somatostatin receptors.

## Data Availability

No original data was produced in the writing of this review article.

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
