# Peer review of "Advances in Molecular Imaging for Neuroendocrine Neoplasms"

_cancers, 2025, doi:10.3390/cancers17122013_

Round 1

Reviewer 1 Report

Comments and Suggestions for Authors

The authors of the review article entitled "Advances in Molecular Imaging for Neuroendocrine Neoplasms" provide a valuable overview of current imaging strategies and emphasise the clinical importance of dual-tracer PET imaging particularly the complementary roles of F-18 FDG and Ga-68 DOTA-TATE in the assessment of neuroendocrine neoplasms. To enhance the clarity, completeness, and clinical utility of the manuscript, I suggest  addressing the following comments:

The authors should consider including a comparative table  or summary outlining how PET/CT imaging, particularly dual-tracer techniques using F-18 FDG and Ga-68 DOTA-TATE can aid in the assessment of tumour differentiation in neuroendocrine neoplasms. Such a table, correlating imaging findings with histological grade (e.g Ki-67 index) and typical CT features, would be a valuable reference for clinicians. It would also assist readers in distinguishing between well-differentiated and poorly differentiated tumours, and in considering differential diagnoses when imaging findings overlap.

2.2. Use of FDG-PET/CT in NENs

Pages 5 and 6, figures 1 and 2: provide hardcoded labels A and B directly onto the images to clearly distinguish between the F-18 FDG PET/CT and the Ga-68 DOTA-TATE PET/CT, respectively.
I would also recommend that all abbreviations such as MIP be spelled out in full at first mention in both the main text and figure legends. Also, it would be beneficial to include key details such as anatomical location of the lesions shown, relevant acquisition parameters, patient positioning, and any segmentation techniques applied.

Provide a brief explanation or legend outlining the basics of scan interpretation. How to distinguish physiologic from pathologic uptake (expected normal distribution of tracer)? This would enhance the educational value and clinical utility of the figures.

Consider specifying the imaging sources and platforms used (scanner model, software, or institutional origin) to support transparency and reproducibility.

Page 5, figure 2: areas of photopenia are noted in image B (Ga-68 DOTA-TATE), it would be helpful for the authors to expand on possible causes such as receptor downregulation, necrosis, prior treatment effect, or technical factors including attenuation correction artefacts or any suboptimal tracer biodistribution.

3. Novel Radiotracers

3.3. Non-SSTR Radiopharmaceuticals 

Page 6,7, table 1: add a clear and descriptive table caption that explicitly defines the column headers, as these are currently missing. Also, to improve the utility and clinical relevance of the table, please consider including additional columns such as radioisotope, key advantages, and clinical impact.

4. Appropriate Patient Selection for Imaging and Follow-up of NEN 

Page 10, line-355:  in relation to the discussion on NETest 2.0, I would encourage the authors to address the current limitation of its interpretability given its nature as a largely algorithm-driven black box model. It would be valuable if the authors could comment on the future potential for integrating imaging features particularly quantitative data or radiomic characteristics derived from CT /PET with molecular profiles obtained from liquid biopsies (e.g. gene expression).

Author Response

The authors of the review article entitled "Advances in Molecular Imaging for Neuroendocrine Neoplasms" provide a valuable overview of current imaging strategies and emphasize the clinical importance of dual-tracer PET imaging particularly the complementary roles of F-18 FDG and Ga-68 DOTA-TATE in the assessment of neuroendocrine neoplasms. To enhance the clarity, completeness, and clinical utility of the manuscript, I suggest addressing the following comments: 

The authors should consider including a comparative table or summary outlining how PET/CT imaging, particularly dual-tracer techniques using F-18 FDG and Ga-68 DOTA-TATE can aid in the assessment of tumour differentiation in neuroendocrine neoplasms. Such a table, correlating imaging findings with histological grade (e.g Ki-67 index) and typical CT features, would be a valuable reference for clinicians. It would also assist readers in distinguishing between well-differentiated and poorly differentiated tumours, and in considering differential diagnoses when imaging findings overlap. Thank you for this very important comment. We appreciate the possibility of improving our article. To address the points raised we have created a figure embedded with a table (Figure 3). We have also added a column to table 1 summarizing the advantages of radiolabels F18, Cu64, and Ga68. 

2.2. Use of FDG-PET/CT in NENs 

Pages 5 and 6, figures 1 and 2: provide hardcoded labels A and B directly onto the images to clearly distinguish between the F-18 FDG PET/CT and the Ga-68 DOTA-TATE PET/CT, respectively. 
I would also recommend that all abbreviations such as MIP be spelled out in full at first mention in both the main text and figure legends. We thank the reviewers for this important suggestion, it will certainly improve readability, hence we have appended the hard labels directly in figure. Also, it would be beneficial to include key details such as anatomical location of the lesions shown, relevant acquisition parameters, patient positioning, and any segmentation techniques applied. As requested by the reviewer we have now provided the anatomical location of the lesions. However, we think acquisition parameters, patient positioning, segmentation techniques are beyond the scope of this review article. 

Provide a brief explanation or legend outlining the basics of scan interpretation. How to distinguish physiologic from pathologic uptake (expected normal distribution of tracer)? This would enhance the educational value and clinical utility of the figures. Consider specifying the imaging sources and platforms used (scanner model, software, or institutional origin) to support transparency and reproducibility. We have modified the legend and hope this will help in improving the readability of the review article. However, we believe that providing scanner model may only be misleading as it will raise the question in the mind of general readers whether the images would have looked differently on other scanner types. 

Page 5, figure 2: areas of photopenia are noted in image B (Ga-68 DOTA-TATE), it would be helpful for the authors to expand on possible causes such as receptor downregulation, necrosis, prior treatment effect, or technical factors including attenuation correction artefacts or any suboptimal tracer biodistribution. We thank the reviewer for the comment. Indeed, we believe that the lack of somatostatin receptor expression on markedly FDG avid hepatic lesions is primarily due to dedifferentiation. Aggressive disease can lose somatostatin receptor (SSTR) expression during natural evolution or can be de-novo SSTR negative.  

  1. Novel Radiotracers

3.3. Non-SSTR Radiopharmaceuticals  

Page 6,7, table 1: add a clear and descriptive table caption that explicitly defines the column headers, as these are currently missing. Also, to improve the utility and clinical relevance of the table, please consider including additional columns such as radioisotope, key advantages, and clinical impact. We are sorry for any problem caused during the process of uploading and conversion of table in pdf. In the word document, all the table headers appear to be okay. 

We have added one more column to define the properties of radioisotopes. 

  1. Appropriate Patient Selection for Imaging and Follow-up of NEN 

Page 10, line-355:  in relation to the discussion on NETest 2.0, I would encourage the authors to address the current limitation of its interpretability given its nature as a largely algorithm-driven black box model. It would be valuable if the authors could comment on the future potential for integrating imaging features particularly quantitative data or radiomic characteristics derived from CT/PET with molecular profiles obtained from liquid biopsies (e.g. gene expression). An additional citation and statements have been added to this section. AI tools are a black box to most clinicians because most are not trained to assess their complex statistical outputs and algorithms associated with AI. We do not think that it is the limitation of the “black box” rather our own (clinicans perspective) limited understanding of how AI works. However, similar to NETest 2.0, there are several other AI based biomarkers currently being explored.  

Reviewer 2 Report

Comments and Suggestions for Authors

Thanks for adding your voice to the NETs topic.

My comments below:

  1. Is surgery rather not the first line of treatment for NETs?
  2. Authors should maybe provide examples of somatostatin analogs for NETs treatment.
  3. Is there any reported known biomarker for NETs that can be seen through urine or blood for example.
  4. How is chromogranin A related to NETs? Can it be considered a biomarker?
  5. PRRT falls under radiotherapy. Authors should rather mention radiotherapy and provide PRRT as an example (line 50).
  6. Somatostatin stabilizes the tumor growth.
  7. Which modality is ideal for NETs, F-18 or Ga-68 based PRRT?
  8. Authors should cite these papers: https://jnm.snmjournals.org/content/64/6/835.long under section 3 of the manuscript and https://www.mdpi.com/1999-4923/13/5/599 under introduction. These reviews talk about radiopharmaceuticals, NETs and their imaging.
  9. Which SSTR is mostly expressed by NETs? I missed it.
  10. Authors highlighted some advantages of agonist and antagonist, but I didn't see which is which. Is agonist better than antagonist? This is an ongoing debate.
  11. Is it safe to say antagonist are better suited for imaging whiles agonist are for therapy?

Author Response

Thanks for adding your voice to the NETs topic.

My comments below:

  1. Is surgery rather not the first line of treatment for NETs? Surgery is the only curative option in NEN. We have modified the sentences and tables to make sure that that no wrong message is sent to the readers.
  2. Authors should maybe provide examples of somatostatin analogs for NETs treatment. Thank you. Although this is not the scope of our manuscript, we have added a sentence on line 48 “Somatostatin analogs such as octreotide and lanreotide are the first line of therapy for the treatment of Grade 1 (G1) and Grade 2 (G2) neuroendocrine tumors (NET).”
  3. Is there any reported known biomarker for NETs that can be seen through urine or blood for example.
  4. How is chromogranin A related to NETs? Can it be considered a biomarker? Question 3 and 4 are important as there are many biomarkers that can be tracked in the urine or blood but deserve a dedicated discussion.
  5. PRRT falls under radiotherapy. Authors should rather mention radiotherapy and provide PRRT as an example (line 50). I introduced PRRT as peptide receptor radionuclide therapy in that part of the paper. Although radiotherapy work by radiation like PRRT, there are important fundamental differences in the mechanism of action that make it distinct. PRRT is a systemic therapy whereas radiotherapy in general is a local-regional therapy. PRRT is affected by many factors such as the tumor sink effect, impact of cold peptides, chelators impact on biodistribution, delayed efficacy (up to 3 years from treatment), delayed side effects, and other such distinctions separate it from radiotherapy.
  6. Somatostatin stabilizes the tumor growth.
  7. Which modality is ideal for NETs, F-18 or Ga-68 based PRRT? The imaging of NETs with F-18, Ga-68, and Cu-64 is often used based on clincial availability, but no specific labeling agent has been determined to be superior to the other in therapeutic decision making.
  8. Authors should cite these papers: https://jnm.snmjournals.org/content/64/6/835.long under section 3 of the manuscript and https://www.mdpi.com/1999-4923/13/5/599 under introduction. These reviews talk about radiopharmaceuticals, NETs and their imaging. These articles have been referenced.
  9. Which SSTR is mostly expressed by NETs? I missed it. An addition has been made to line 90 and this was also mentioned in the body of the paper but an important point. Line 90 “These SSTRs, particularly SSTR2 expressed in neuroendocrine tumors can be visualized non-invasively by labeling artificially synthesized somatostatin analogs with positron emitters such as F-18, Ga-68, or Cu-64.”
  10. Authors highlighted some advantages of agonist and antagonist, but I didn't see which is which. Is agonist better than antagonist? This is an ongoing debate. We have deliberately not taken the agonist vs antagonist in this review as it is beyond the scope of the article. In addition, there is no study that has shown that an antagonist can lead to a significant change in therapeutic management.
  11. Is it safe to say antagonist are better suited for imaging whiles agonist are for therapy? I agree that questions 10 and 11 are important but I would rather not speculate at this time. It is possible that antagonists will be highly effective in therapy, although comparative data for therapy is not available.

Round 2

Reviewer 1 Report

Comments and Suggestions for Authors

The authors have thoroughly revised the manuscript and addressed all comments and suggestions; however, Table 2 on page 14 should be corrected, as its caption appears beneath the table, which is inappropriate if it is indeed a real table rather than a figure.

Author Response

The authors have thoroughly revised the manuscript and addressed all comments and suggestions; however, Table 2 on page 14 should be corrected, as its caption appears beneath the table, which is inappropriate if it is indeed a real table rather than a figure.

Thank you for drawing attention to this. The image in question is more appropriately a figure and that correction has been made.